# An Exploratory Analysis of the Association between Hospital Quality Measures and Financial Performance

**DOI:** 10.3390/healthcare11202758

**Published:** 2023-10-18

**Authors:** Brad Beauvais, Diane Dolezel, Zo Ramamonjiarivelo

**Affiliations:** 1School of Health Administration, Texas State University, Encino Hall, Room 250A, 601 University Drive, San Marcos, TX 78666, USA; zhr3@txstate.edu; 2Health Informatics & Information Management Department, Texas State University, Round Rock, TX 78665, USA; dd30@txstate.edu

**Keywords:** hospital, quality, patient safety, perceptions of care, financial performance

## Abstract

Hospitals are perpetually challenged by concurrently improving the quality of healthcare and maintaining financial solvency. Both issues are among the top concerns for hospital executives across the United States, yet some have questioned if the efforts to enhance quality are financially sustainable. Thus, the aim of this study is to examine if efforts to improve quality in the hospital setting have a corresponding association with hospital profitability. Recent and directly relevant research on this topic is very limited, leaving practitioners uncertain about the wisdom of their investments in interventions which enhance quality and patient safety. We assessed if eight different quality measures were associated with our targeted measure of hospital profitability: the net patient revenue per adjusted discharge. Using multivariate regression, we found that improving quality was significantly associated with our targeted measure of hospital profitability: the net patient revenue per adjusted discharge. Significant findings were reported for seven of eight quality measures tested, including the HCAHPS Summary Star Rating (*p* < 0.001), Hospital Compare Overall Rating (*p* < 0.001), All-Cause Hospital-Wide Readmission Rate (*p* < 0.01), Total Performance Score (*p* < 0.001), Safety Domain Score (*p* < 0.01), Person and Community Engagement Domain Score (*p* < 0.001), and the Efficiency and Cost Reduction Score (*p* < 0.001). Failing to address quality and patient safety issues is costly for US hospitals. We believe our findings support the premise that increased attention to the quality of care delivered as well as patients’ perceptions of care may allow hospitals to accentuate profitability and advance a hospital’s financial position.

## 1. Introduction

### 1.1. Background

Healthcare spending, as a percentage of Gross Domestic Product (GDP), is the highest in the United States (US) compared to any other country in the world. As of 2021, the percent of GDP spent on health in the US stood at 17.8%, far higher than the next highest group of developed nations, including Germany (12.8%), France (12.4%), Great Britain (11.9%), and Canada (11.7%) [1]. However, the US has a higher mortality rate for treatable or preventable conditions and double the mortality rate for multiple chronic conditions when compared to other countries [1]. This indicates that increased spending may not be directly related to higher quality of care or higher patient outcomes. According to the Centers for Disease Control and Prevention (CDC), the costs associated with chronic diseases and mental health treatments accounted for 90% of all annual US healthcare expenses [2]. In 2021, spending on healthcare in the U.S. increased to USD 4.3 trillion (about USD 13,000 per person in the U.S.) [3]. Hospital care accounted for 31.1% of the 2021 expenditures, and private health insurance (28.5%), Medicare (21.2%), and Medicaid (17.2%) were the largest payers [4]. 

Partially because of these national trends, patient safety and quality of care concerns have become more prominent in the past two decades. The Institute of Medicine’s (IOM’s) 2000 seminal report, *To Err Is Human: Building a Safer Health System,* estimated that as many as 98,000 deaths in hospitals each year could be the result of inpatient safety issues [5]. A more recent study indicates that medical errors are the third leading cause of death in the U.S. and reported that preventable medical errors surpass 250,000 annually [6]. In response to these issues, numerous public and private entities have sought to advance patient safety and quality-enhancing initiatives. For example, in 2005, the Center for Medicare and Medicaid Services (CMS) pay-for-performance initiative associated Hospital-Acquired Conditions (HACs) with reduced Medicare payments as an incentive to reduce infection rates and improve overall patient safety [7]. In 2006, the Hospital Consumer Assessment of Healthcare Providers and Systems (HCAHPS) surveys of discharged patients were nationally implemented and included questions on several patient experience measures, including patient-provider communications, the hospital environment, discharge instructions, and more. By 2007, the Institute for Healthcare Improvement’s (IHI) Triple Aim of Healthcare framework for improving health system performance called for the concurrent pursuit of improving the patient care experience, improving the health of populations, and reducing the per capita costs of healthcare [8]. Similarly, the Patient Protection Affordable Care Act of 2010 created the Hospital Value-Based Purchasing Program (VBP), and the Medicare Access and Children’s Health Insurance Program (CHIP) Reauthorization Act of 2015 provided financial rewards to doctors via the Quality Payment Program [9]. Both are focused on improving the quality of care for Medicare patients [10,11]. And, earlier this year, the Joint Commission’s National Patient Safety Goals for 2023 suggestions for improving patient safety included the use of medicines safely and preventing mistakes in surgery [12]. 

### 1.2. Relevance

These broad and expansive sets of guidance, regulations, and policies from numerous agencies and sources put hospitals and healthcare systems under increasing pressure to provide higher quality services, concurrently constrain costs for patient care, and maintain financial viability. This can be a challenging endeavor to accomplish. The American College of Healthcare Executives 2022 annual survey of Chief Executive Officers discloses that ‘financial challenges’ of healthcare organizations are a continuing top concern (#2), along with ‘patient safety and quality’ (#4), ‘governmental mandates’ (#5), and ‘patient satisfaction’ (#7) [13,14]. In particular, rural hospitals’ financial solvency has been a persistent issue [15]. A few subspecialties have managed to increase healthcare quality while controlling costs, but efforts to improve the value of healthcare systems by reducing costs, improving quality, and improving patient outcomes remain inconsistent across the US healthcare systems [16]. 

Barriers to maintaining financial viability in the healthcare industry are numerous. They include payer reimbursement tied to quality-of-care improvements, patient safety mandates, cost of new technologies, unplanned readmissions, supply-driven demand, and the increasing costs of healthcare delivery (e.g., labor, telehealth, medical devices, and supply chain expenses) [17]. However, failing to address patient safety issues is costly for US hospitals. As just one example, the cost of treating preventable adverse drug events related to inappropriately overriding medicine alerts was estimated to be between USD 871 million and 1.8 billion for US hospitals [18]. And, when considered collectively, adverse events account for an estimated cost of USD 20 billion annually in the United States [19]. 

The prevalence and cost of preventable adverse events can affect hospital quality ratings and profitability. Related research has found positive quality–profitability relationships in numerous non-healthcare industries [20,21,22,23]. However, such research is relatively sparse in the healthcare setting. Barnes et al. (2017) only found 13 studies assessing the association between hospital quality measures and financial performance over 20 years (1997–2017) in their systematic literature review [24]. Bai and Anderson (2016) suggest examining the quality factors related to hospital profitability, but they did not include this in their study on the topic [25]. Similarly, Holt et al. (2011) and Enumah et al. (2022) also encourage the future examination of the association of quality-related organizational factors and hospital financial performance [23,26]. Although informative in their guidance, these few studies leave several unanswered questions to resolve. None of the studies incorporate the present measures of patient perceptions of quality, nor do they include measures that are currently financially incentivized by CMS and directly tied to organizational improvement. 

Thus, building on the work of the referenced studies and noting their recognized limitations, the research team seeks to advance our understanding of the association between several established quality measures and hospital profitability in acute care hospitals in the United States. We specifically aim to assess whether the quality of care from the perspective of the patient and clinical data is associated with a hospital’s ability to attract patients as measured by revenue. We intend to provide healthcare leaders and policy developers with insight into the studied relationship in the context of the current market and hospital characteristics.

## 2. Literature Review 

### 2.1. Hospital Financial Performance

Although there is a general lack of studies that evaluate the association between quality and financial performance in the healthcare setting, there are several prior studies that have examined the factors directly associated with financial performance. Such factors have included numerous organizational factors, practices, strategies, and/or organizational profiles. With respect to the relationship between organizational practices and/or strategies and financial performance, Kaissi et al. (2008) found that having a strategic plan, having the CEO lead the strategic planning regimen, and involving the Board of Directors are all factors that are positively associated with financial performance in terms of net income and profit margin [27]. Another study regarding hospital–physician integration strategies found that involving physicians in hospital governance was associated with higher operating margins, and the integration of financial arrangements between the hospitals and physicians was associated with decreased cost, while direct hospital ownership among physicians was associated with decreased operating margins and increased cost [28]. A structural equation modeling study on the relationships between medical technology and information technology adoption strategies and financial performance found that both technologies were positively associated with financial performance in terms of the composite measure of return on assets, return on investment, and operating profit [29]. In the same vein, the study of Zengul et al. (2018) found that the breadth of high-technology services was positively associated with total margin, among for-profit hospitals, while both the breadth and rareness of high medical technology were positively associated with total margin among not-for-profit hospitals [30]. The study on the adoption of patient engagement strategy found that early adopters exhibited higher operating margins compared to the majority and later adopters [31]. Nurse staffing strategy (increased nurse-to-patient ratio) was also found to be associated with increased hospital total profit margin, especially for hospitals operating in competitive markets, while McCue, Markm, and Harless (2003) found that registered nurse staffing level was positively associated with operating cost but had no significant effect on profit [32,33]. The more recent study by Lee et al. (2022) showed that hiring more advanced practice registered nurses compared to physicians was positively associated with operating margin and return on assets [34]. Another staffing strategy regarding hospitalists indicated that using a high hospitalist staffing intensity strategy was associated with both increased revenue and higher operating costs, resulting in a marginally significant increase in operating profit (Epane et al., 2019) [35]. Other organizational strategies such as privatization (Ramamonjiarivelo et al., 2018) and merger (Groff, Lien, Su, 2007) were also found to positively impact financial performance in terms of margins and efficiency, respectively [36,37].

Extant studies examining the association between organizational factors and financial performance indicated that some factors affect financial performance. For instance, Gapenski, Vogel, and Langland-Orban (1993) found teaching status, number of beds, ownership status, system affiliation status, age of plant, case mix, average length of stay, and others [38]. They categorized the determinant factors as organizational, managerial, patient mix, and market variables. Holt et al. (2011) added to the literature by examining current studies and categorizing ownership, governance, management strategy, integration, and quality as the five most studied determinants [26]. Turner, Broom, Elliott, and Lee (2015) looked at the determinants of hospital profitability using the DuPont analysis tool and recommended further study of the relationship between quality outcomes and profitability [39]. Bai and Anderson (2016) used similar metrics in their study of the key factors that most financially successful hospitals share [25]. They found that for-profit ownership, higher markup, regional power, and price regulation had the largest positive association with hospital profitability, and yet they identified quality performance as an area for further study. The more recent study by Lee et al. (2022) found that metropolitan location was positively associated with hospitals’ profitability in terms of return on assets and return on equity, while rural location was associated with loss of profitability (negative operating margin) [34]. They also found that government-owned hospitals and private not-for-profit hospitals are less profitable compared with for-profit hospitals; compared to non-teaching hospitals, teaching hospitals are less profitable in terms of operating margin.

### 2.2. Hospital Quality of Care

The World Health Organization (WHO) defines quality of care as “the degree to which health services for individuals and populations increase the likelihood of desired health outcome” [40]. The Institute of Medicine reports elevated healthcare quality concerns on a national level [41]. Subsequent efforts by numerous organizations including the Centers for Medicare and Medicaid (CMS), the Centers for Disease Control (CDC), and The Joint Commission (TJC) yielded an evolving list of programs designed to improve quality by prompting enhanced organizational performance. The Affordable Care Act of 2010 established the Hospital Value-Based Purchasing Program (VBP), which rewards more than 3000 acute care hospitals with incentive payments for the quality of care they provide to Medicare beneficiaries. Numerous factors combine to influence any acute care hospital’s VBP score [10]. 

The HVBP program continues to evolve on an annual basis. Numerous data elements have been included in the program since its inception [11]. The HVBP program now includes an evaluation of the patient experience of care using the Hospital Consumer Assessment of Healthcare Providers and Systems (HCAHPS) Survey, numerous clinical outcomes, and patient safety indicators (e.g., Central Line-Associated Blood Stream Infection (CLASBI), Catheter-Associated Urinary Tract Infection (CAUTI), Methicillin-Resistant Staphylococcus Aureus (MRSA), pneumonia, heart failure, acute myocardial infarction) as well as efficiency as determined by Medicare Spending per Beneficiary (MSPB). These measures join other CMS value-based programs, including the Hospital Readmissions Reduction Program (HRRP), the Hospital-Acquired Condition Reduction Program (HAC), the Physician Value-Based Modifier Program (PVBM), and the Medicare Access and CHIP Reauthorization Act of 2015 (MACRA) Quality Payment Program. 

### 2.3. Hospital Quality of Care and Financial Performance

Historically, hospital leaders have been reluctant to invest in quality improvement and safety programs, with up to 92% indicating they did not have a budget line item attributed to patient safety twenty years ago [41]. Some authors have noted that not all quality improvement initiatives are economically sustainable or provide a return to the organization [42]. Despite the prevalence of improvement programs in the United States, the evidence is mixed regarding whether these programs have had a positive effect [43,44,45,46]. Some have expressed concern about whether the efforts invested in quality improvement have had any meaningful return on investment [47]. Others have questioned whether the incentives in the current value-based purchasing models are sufficient to impart consequential improvement and if the aggregate results are worth the efforts involved in improving performance [48,49].

To address this issue, in a limited number of studies, some have endeavored to investigate the ‘business case’ of quality in the healthcare industry. Harkey and Vraciu (1992) and Alexander, Weiner, and Griffith (2006) both reported hospitals that pursue broad and intense baseline quality improvement programs demonstrated improved financial performance [50,51]. Similarly, Nelson et al. (1992) in their study of 15,095 randomly selected patients from 51 general medical/surgical hospitals owned by the Hospitals Corporation of America (HCA) found that patient ratings of hospital services, based on a validated and reliable survey instrument (Hospital Quality Trends: Patient Judgment System), were positively associated with earnings before depreciation, interest and taxes per bed, net revenue per bed, and return on assets [52]. In the same vein, Velez-Gonzalez et al. (2011) assessed the association between non-financial performance measures and financial performance of for-profit system hospitals, and their finding suggest that quality of care in terms of Joint Commission’s quality composite score was positively associated with total margin and operating margin [53]. 

More recent studies focused on a limited set of outcome measures or adverse events and their associated financial implications. As an example, Beauvais, Richter, and Kim (2019) examined the impact of hospital safety scores from the Leapfrog Group on hospital financial performance and concluded that there was a positive relationship between the scores and all the financial measures tested [54]. In subsequent work, Beauvais et al. (2019) also determined that there is a positive association between hospital patient safety measures and financial performance [55].

## 3. Methods

### 3.1. Conceptual Framework and Hypotheses

W. Edward Deming’s statement, “Profits are the result of attention to quality and customer satisfaction, while the reverse is rarely true”, captures a viewpoint that has reformed numerous industries since the 1950s [56]. We refer to Deming’s insights for our research and draw further clarity from the later work of Rust, Zahorik, and Keiningham (1995), who provide a guide to conceptualizing how improved quality performance can simultaneously increase revenues, reduce costs, enhance market share, and positively influence profitability in the service industry [57]. These authors indicate superior service quality supports profitability in the services sector in two ways. First, the direct impact of cost reductions generated from service quality improvement on profitability. Second, quality improvement efforts can indirectly boost revenues via improved customer perception of quality, customer satisfaction, and customer retention. Third, positive word-of-mouth from satisfied customers attracts new customers, which ultimately increases revenues and market share. 

When considered in its entirety, the body of empirical literature guides us to conjecture that quality-enhancing initiatives at the hospital level are likely to be associated with improved financial outcomes. Therefore, we hypothesize that:

**Hypothesis** **1 (H1).**
*Hospitals with better patient perceptions of quality performance will be associated with improved financial performance.*


**Hypothesis** **2 (H2).**
*Hospitals with lower readmission rates performance will be associated with improved financial performance.*


**Hypothesis** **3 (H3).**
*Hospitals with better HVBP Total Performance Scores will be associated with improved financial performance.*


**Hypothesis** **4 (H4).**
*Hospitals with better HVBP Domain Scores will be associated with improved financial performance.*


### 3.2. Data

Data were obtained from Definitive Healthcare which contains the databases of several US healthcare organizations, such as hospitals, physician group practices, surgery centers, and long-term care organizations [58]. Concerning US hospital data, Definitive Healthcare combines data from several sources, such as the American Hospital Association Annual Survey (hospital profile), Medicare Cost Report (financial data), the Hospital Value-Based Purchasing Program (quality data), and Hospital Compare (quality data). Definitive Healthcare provided 2127 hospital observations for the year 2022. The original dataset consisted of all 3876 short-term acute care hospitals in the United States. All Federal hospitals, including 172 Veterans Affairs, 26 Indian Health Service, and 31 Military Health System facilities, were excluded from our study sample due to a lack of numerous relevant data elements. We removed an additional 1520 facilities because of significant data missingness—particularly in the independent variables of interest. The final dataset comprises 54.8 percent of the total active short-term acute care facility population in the United States. 

### 3.3. Dependent Variable

Consistent with the Bai & Anderson (2016) study about the determinants of hospital profitability, the dependent variable analyzed in our study was the hospital’s net patient revenue per discharge (NPRPD) for the year 2022 [25]. The year 2022 was specifically chosen as it was the most recent complete year of data. “Net patient revenue” reflects revenue for patient care only and does not include revenue from other operations such as the cafeteria, parking, rent, research, and educational activities. A “discharge” is defined as the formal release of a treated individual due to the conclusion of the clinical stay, either by death, return home, or transfer to another institution. 

### 3.4. Independent Variables of Interest

Numerous quality assessment measures of acute care hospital operations are available from various sources. This facilitated our analysis and allowed us to choose a diverse set of independent variables drawn from well-established and publicly available data. The first independent variable includes the 2021 HCAHPS (Hospital Consumer Assessment of Healthcare Providers and Systems) Summary Star Rating [57]. The HCAHPS survey asks discharged patients 29 questions about their experience with a hospital stay, including questions about communication with nurses and doctors, the responsiveness of hospital staff, the cleanliness and quietness of the hospital environment, communication about medicines, discharge information, overall rating of the hospital, and would they recommend the hospital. The patient survey summary star rating is the average of all the Star Ratings of the HCAHPS measures. Hospitals can earn 1 to 5 stars for this metric, in which more stars are better [59]. The findings from the evaluation of the HCAHPS variable were used to support the evaluation of Hypothesis 1.

The second independent variable included in the study is the 2021 Hospital Compare Overall Rating. This measure provides consumer-focused aggregated scores related to hospitals’ performance by taking the weighted average of scores calculated based on measures of mortality, safety of care, readmission, patient experience, effectiveness of care, timeliness of care, and efficient use of medical imaging. The Hospital Compare Overall Rating is calculated using only measures reported to CMS through the Hospital Inpatient Quality Reporting (IQR) and Hospital Outpatient Quality Reporting (OQR) Programs. Hospitals can earn 1 to 5 stars for this metric, in which more stars are better [60]. The findings from the evaluation of the Hospital Compare Overall Rating variable were used to support the evaluation of Hypothesis 1.

The third independent variable was the 2021 All Cause Hospital-Wide Readmission Rate. The 30-day rate indicates how many patients had to be readmitted back into a hospital within 30 days after they were originally discharged. Hospitals maintain lower readmission rates when they have appropriately resolved the patient’s healthcare needs without further intervention [61]. The findings from the evaluation of the All-Cause Hospital-Wide Readmission Rate variable were used to consider Hypothesis 2.

The fourth through eighth variables considered included the 2021 Hospital Value-Based Purchasing (VBP) Total Performance Score and associated domain scores. Value-based purchasing is a CMS program that adjusts a hospital’s payments based on its performance in four equally weighted quality measurement domains to comprise its Total Performance Score. The domains include (1) clinical outcomes, (2) safety, (3) person and community engagement, and (4) efficiency and cost reduction [62]. The “clinical outcomes” domain contains measures six measures, including mortality from acute myocardial infarction (AMI), chronic obstructive pulmonary disease (COPD), coronary artery bypass graft (CABG), heart failure (HF), pneumonia (PN), and measure of complications from elective primary total hip/total knee arthroplasty. The “safety” domain contains six healthcare-associated infection measures, including catheter-associated urinary tract infections (CAUTIs), central line-associated bloodstream infections (CLABSIs), clostridium difficile infections (C. diff), methicillin-resistant staphylococcus aureus bacteremia (MRSA), surgical site infection (SSI) from abdominal hysterectomy, and SSI from colon surgery. The “person and community engagement” domain contains eight dimensions derived from the HCAHPS Survey, including communication with nurses, communication with doctors, responsiveness of the hospital staff, communication about medicines, cleanliness and quietness of the hospital environment, discharge information, care transitions, and the overall rating of the hospital. Lastly, the “efficiency and cost reduction” domain contains one measure related to Medicare Spending per Beneficiary [62]. The findings from the evaluation of the HVBP Total Performance Score variable were used to test Hypothesis 3, while the study of all four HVBP sub-domain scores was used to test Hypothesis 4.

### 3.5. Controls

Numerous independent variables are included in the study to account for the variation in hospital profitability associated with various individual hospitals and hospital market characteristics, including the total assets per staffed bed (in millions), the complication/comorbid and major complication/comorbid (CC/MCC) rate, urban or rural location, local hospital market concentration (as measured via the Herfindahl–Hirschman Index), government-operated or not, the average daily census, surgical case mix index, medical case mix index, and overall case mix index, Medicaid days of service, Medicare days of service, bed utilization rate, average age of the facility (in years), average length of stay, amount of uncompensated care (in millions), amount of charity care (in millions), the labor compensation ratio, and geographic region of the country (Southeast, Southwest, Midwest, West, or Northeast).

### 3.6. Analysis

The potential for reverse causality prompted us to use older quality data from the various public datasets to ensure that our two datasets did not fully overlap. This allows for the impact of improved quality performance to be realized in the hospital financial reporting systems. The practice of replacing an explanatory variable with its lagged value to counteract endogeneity is prevalent across a wide variety of disciplines in economics and finance [54,63,64]. Due to the skewness of the dependent variable, the distribution was shifted, and natural log transformed. Due to the natural log transformation of the dependent variables, interpretation of our results requires adjustment of the parameter estimates for final analysis; we would say that an increase of one unit in 𝑥 is associated with a 100 × (𝑒^𝛽^ − 1) change in 𝑦. In more simple terms, this implies there is a percent change in y associated with a one-unit increase in x.

Multicollinearity was evaluated, and any variables with a variance inflation factor over 10 were removed. To aid in ease of interpretation, all independent variables of interest were treated as continuous variables, including the Likert-scale Hospital Compare and HCAHPS Star Ratings. This approach is in alignment with prior research that indicates ordinal variables with five or more categories can be used as continuous data without any harm to the analysis [65,66,67,68,69]. Several control variables were also included as dichotomous measures, including rural = 1, urban = 0; government = 1, not government operated = 0; and for-profit = 1, not-for-profit = 0. Eight multiple linear regressions with listwise deletion were conducted using IBM (International Business Machines) SPSS (Statistical Package for Social Sciences) Statistics package 28 [69]. In each of the analyses performed, the association between the studied independent variables and the dependent variable was rejected at an α = 0.05. Model fit was assessed using adjusted *R*^2^.

## 4. Results

A descriptive analysis of all variables is available in Table 1. Our sample is comprised of only 8% rural hospitals (SD = 0.27), of which 13% are government operated (SD = 0.34), 20% are for-profit (SD = 0.40), and the majority of which are hospitals in the Southeast Region (30%; SD = 0.46). Our sample hospitals maintain an average age of plant of just over 14 years (SD = 10.17), keep an average of USD 2.45 million in assets per staffed bed (SD = 6.34), sustain USD 26.73 million in uncompensated care (SD = 56.69), perform USD10.11 million in charity care each year (SD = 26.0), experience a CC/MCC rate of 67% on average (SD = 0.07), and manage an average daily census of 146.36 (SD = 169.94). On average, these facilities’ patients are comprised of 9% Medicaid (SD = 0.08) and 28% Medicare (SD = 0.10) patients, utilize 56% bed occupancy (SD = 0.19), and utilize 45% of revenue to compensate their employees (SD = 0.19). 

### 4.1. Primary Findings

Table 2 presents the multivariate regression results for our first four independent variables of interest. The beta coefficients, standard error (S.E.), and significance (Sig.) are given for the HCAHPS rating and Hospital Compare rating, Readmission Rate, and Total Performance Score. Our regression findings indicate that quality is associated with higher levels of hospital profitability across each of the first four quality dimensions. In our first analysis, hospital HCAHPS Summary Star Rating is positively associated with Net Patient Revenue per Discharge (*R*^2^ = 48.2%, *β*: 0.088, S.E.: 0.01, *p* < 0.001). One practical interpretation of these results, given that our dependent variable is natural log transformed, is that a one-point increase in a hospital’s HCAHPS Summary Star rating is associated with an 8.8% increase in Net Patient Revenue per Discharge. These findings are supportive of our first hypothesis (H1). Similar findings were observed in the analysis of the Hospital Compare Rating. In this analysis, the Hospital Compare rating is positively associated with Net Patient Revenue per Discharge (*R*^2^ = 45.2%, *β*: 0.034, S.E.: 0.01, *p* < 0.001). This could be interpreted to mean that a one-point increase in a hospital’s Hospital Compare rating is associated with a 3.4% increase in Net Patient Revenue per Discharge. These findings also are supportive of our first hypothesis (H1).

With respect to hospitals’ All-Cause Readmission Rate, we observed a negative association with Net Patient Revenue per Discharge (*R*^2^ = 43.9%, *β*: −0.017, S.E.: 0.01, *p* < 0.01). We could infer this to mean that a 1% increase in the All-Cause Readmission Rate is associated with a 1.7% decrease in Net Patient Revenue per Discharge. These findings are supportive of our second hypothesis (H2). In the final column in Table 2, we noted that the HVBP Total Performance Score is positively associated with Net Patient Revenue per Discharge (*R*^2^ = 45.8%, *β*: 0.005, S.E.: 0.00, *p* < 0.001). This implies that for a one-point increase in Total Performance Score, there is an associated 0.5% increase in Net Patient Revenue per Discharge. These findings are supportive of our third hypothesis (H3).

Table 3 presents similar multivariate regression results for our second set of four independent variables of interest pertaining to the HVBP sub-domains. In this set of variables, our findings still generally indicate that quality is associated with higher levels of hospital profitability, with one exception. In our first analysis in Table 3, the HVBP Clinical Domain score is not significant. This finding is contrary to our fourth hypothesis (H4).

In our second analysis in Table 3, we note that the HVBP Safety Domain Score is positively associated with Net Patient Revenue per Discharge (*R*^2^ = 46.2%, *β*: 0.003, S.E.: 0.001, *p* < 0.01). One practical interpretation of these results is that a one-point increase in a hospital’s Safety Domain Score is associated with a 0.3% increase in Net Patient Revenue per Discharge. These findings are supportive of our fourth hypothesis (H4).

In our third analysis, we observed that the HVBP Engagement Domain Score is positively associated with Net Patient Revenue per Discharge (*R*^2^ = 48.6%, *β*: 0.016, S.E.: 0.001, *p* < 0.001). This could be interpreted to mean that a one-point increase in a hospital’s Engagement Domain Score is associated with a 1.6% increase in Net Patient Revenue per Discharge. These findings are supportive of our fourth hypothesis (H4).

In our final analysis in Table 3, we see a positive association between the HVBP Efficiency Domain Score and Net Patient Revenue per Discharge (*R*^2^ = 46.3%, *β*: 0.008, S.E.: 0.001, *p* < 0.001). This could be interpreted to mean that a one-point increase in a hospital’s Efficiency Domain Score is associated with a 0.8% increase in Net Patient Revenue per Discharge. These findings are also supportive of our fourth hypothesis (H4).

### 4.2. Secondary Findings

Although not originally considered in our hypothesis testing, there were several interesting secondary findings related to hospital net patient revenue per discharge that are worth noting in our analysis. Although there was variation across the eight analyses, the findings were relatively consistent, and some were surprising in their directionality. As a sample of the findings, we will use the HCAHPS Summary Rating results from Table 2 as our reported analysis in this section. As such, we noted positive associations with NPRPD from total assets per staffed bed (+1.2% in NPRPD per million in assets; S.E.: 0.001, *p* < 0.001), the CC/MCC rate (+36.3% in NPRPD per point change; S.E.: 0.098, *p* < 0.001), rural facilities (+6.4% in NPRPD; S.E.: 0.025, *p* < 0.001), market concentration (+13.3% in NPRPD per point increase; S.E.: 0.018, *p* < 0.001), and government-operated facilities (+7.1% in NPRPD; S.E.: 0.017, *p* < 0.001). Similar positive associations were noted with the overall case mix index (+42.4% in NPRPD per point increase; S.E.: 0.032, *p* < 0.001), Medicare days (+0.2% in NPRPD per day; S.E.: 0.001, *p* < 0.001), average length of stay (+8.7% in NPRPD per day; S.E.: 0.006, *p* < 0.001), and the Western geographic region (+10.3% in NPRPD—compared to the Northeast region; S.E.: 0.019, *p* < 0.001). 

Significant negative associations were also noted, including for-profit status (−8.8% in NPRPD per day; S.E.: 0.014, *p* < 0.001), surgical case mix index (−18.5% in NPRPD per unit increase; S.E.: 0.018, *p* < 0.001), medical case mix index (−24.3% in NPRPD per unit increase; S.E.: 0.076, *p* < 0.001), bed utilization (−0.3% in NPRPD per point increase; S.E.: 0.000, *p* < 0.001), average age of facility (−0.1% in NPRPD per year increase; S.E.: 0.001, *p* < 0.01), charity care (−0.1% in NPRPD per million; S.E.: 0.000, *p* < 0.05), and labor compensation ratio (−0.2% in NPRPD per percent increase; S.E.: 0.000, *p* < 0.001). Both the Southeast geographic region (−16.2% in NPRPD; S.E.: 0.017, *p* < 0.001) and Southwest geographic region (−15.7% in NPRPD; S.E.: 0.021, *p* < 0.001) reflected a significantly lower NPRPD when compared to the Northeast region. 

## 5. Discussion

In general, our results indicate that improved quality performance is associated with improved hospital profitability, as measured by the net operating margin per discharge. Apart from the HVBP clinical care sub-domain, our findings appear to support the prior work of Rust, Zahorik, and Keiningham (1995), who conceptualized quality as a two-pronged value to the organization via market recognition and improved internal performance [55]. As we examine our study outcomes in more detail, we first turn our attention to the HVBP Total Performance Score and note the strong association that exists between this measure of hospital performance and net patient revenue per discharge. To extrapolate the results further, with each additional point in the TPS score, measured on a 0–100 scale, our results indicate that there is an associated 0.5% increase in net patient revenue per discharge. This would seem to indicate that improving overall organizational quality is at least partially self-sustaining financially. 

Our findings related to HCAHPS, the Hospital Compare Star Rating, and HVBP Engagement scores lend further support to the premise that quality performance is associated with revenue enhancement. These results, each measuring a slightly different perspective of patient perceptions of quality, indicate that improved quality is recognized in the marketplace by payers, providers, and patients, which may have a downstream impact on profitability. In addition, we observe that quality improvements related to readmission rates and safety performance are also associated with improved financial position. This appears to support the work of both Beauvais, Richter, and Kim (2019) and Beauvais et al. (2019) [52,53]. In each case, the authors were able to identify a statistically significant association between patient safety performance and improved profitability. Logically, the readmissions finding is related to the fact hospitals are liable for the total expense of readmissions resulting from a prior episode of care within the past 30 days. It is also plausible to consider improved safety scores related to performing care with fewer costly errors, less waste, and increased efficiency. 

In contrast to our collective findings showing an association between quality and financial performance, we noted some relationships in our analysis that require additional thought. Among our primary independent variables of interest, only the HVBP clinical performance sub-domain was found to not be significantly associated with hospital profitability. This is ironic given the central role that clinical care has in generating income for any hospital. However, we conjecture that a more nuanced analysis of the impact of clinical performance is required to gain an appreciation of why the clinical care measure was not associated with profitability. Future researchers might consider evaluating the quality of care or performing a cost analysis at the service line or even the procedural level. Unfortunately, developing a study of this granular nature is not possible with the currently available dataset. 

Our secondary findings are also of interest and worth discussing. Although these results are not our original focus, the magnitude, directionality, and significance of some of the findings are worth noting. The case mix index and CC/MCC rates are perhaps the two most notable variables on this list. The case mix index is a metric that reflects the diversity, complexity, and severity of the patients treated at a healthcare facility, such as a hospital. The case mix index is used by the Centers for Medicare and Medicaid Services (CMSs) to determine hospital reimbursement rates for Medicare and Medicaid beneficiaries [42]. The CC/MCC Rate is a measure of the incidence of complications (CCs) and major complications (MCCs) within a period. The numerator is the number of patients with Medicare Severity Diagnosis-Related Groups (MS-DRGs) defined by the presence of a CC or MCC. The denominator is the total number of Inpatient Prospective Service (IPPS) patients [42]. In our study, we find that a full point change in CMI (range 1.01–3.85) is associated with a 42.4% increase in NPRPD. Likewise, a full point increase in CC/MCC (range 0.13–0.86) is associated with a 36.3% increase in NPRPD (S.E.: 0.098, *p* < 0.001), but given this variable range, a full point change is not possible. Nonetheless, a tenth of a point is associated with a 3.63% increase, which is a noteworthy increase. Beyond these two measures, several other variables provided intriguing results and may be worthy of future research. 

### Practice Implications

Others have previously suggested that improved hospital quality (e.g., surgical care improvement, safety scores, etc.) was associated with improved financial performance [40,41]. Prior researchers exposed some of the interactions between quality and profitability in hospital systems. However, in this study, we sought to take a more expansive look into the organizational data to see if other quality metrics are associated with financial performance in the same way prior researchers have found with a narrower look at strictly patient safety performance. And, overall, our results were highly consistent within our study and with prior authors’ findings. Specifically, we can infer from our findings that many of the steps taken to improve patient perceptions of quality and safety, reduce readmissions, and improve efficiency are all in the best interest of the patient but also serve to financially support the organization’s long-term economic viability. Although we believe more work needs to be conducted in this area, we contend that our findings continue to add to the evidence base that affirms the business case for improving organizational quality. With our results, healthcare leaders’ support of focused investments in quality and efficiency can be profitable irrespective of the influence of regulatory or value-based purchasing initiatives. 

## 6. Limitations and Suggestions for Future Research

Several limitations are present in our study. First, the current study is drawn from a single data year (2022), and we have lagged independent and control variables (2021) to address endogeneity and reverse causality. In the future, we could consider the tested relationship by using longitudinal data and/or incorporating a more robust dataset with more completed data. Although the current dataset constitutes over half of acute care hospitals in the United States, our results might be altered if the data were more comprehensive and complete for a larger number of hospitals. 

Second, there may be additional significant factors that influence the variation in our chosen dependent variable. Although we tested a broad range of variables in our tested models, and our regressions maintained relatively strong *R*^2^ values, we recognize other variables that influence our studied relationship that we are not capturing in our study. For instance, we conjecture variables such as the service mix, the demographics of the supported patient population, the range of services offered by the hospital, and the composition of the clinical staff may all be relevant in teasing out additional variation in the dependent variable. The inclusion of interaction terms among our studied variables might also be an interesting addition to the research.

A final limitation centers on the fact that all our chosen quality-dependent variables are weighted aggregates. Although this provides consistency across our studied population, additional insight might be gained by examining our study relationship on more granular aspects of each of the current independent variables. As an example, even though we have evaluated each of the sub-domains of the value-based purchasing Total Performance Score, we could delve more deeply into the component measures of the sub-domains. Likewise, future analysis could examine the granular components of the All-Cause Readmission Rate to see if there are specific readmission types that influence hospital profitability. Similar efforts could be applied to the measures supporting the Hospital Compare overall rating or the HCAHPS summary star rating.

## 7. Conclusions

Historically hospital leaders have been reluctant to invest in quality improvement and safety programs. We contend those days are in the past. There has been, and likely always will be, a perceived trade-off between quality improvement and hospital profitability. To our knowledge, this association has never been studied in granular detail with a contemporary set of quality and safety measures. So, in our research, we called the quality–profitability tradeoff premise into question and examined the association between eight unique healthcare quality variables and hospital profitability. Based on our research, we can confidently indicate that the effort expended to enhance the patient experience, reduce readmissions, and improve patient safety is associated with improved hospital financial performance. Given the expanding role of value-based case incentives and evolving healthcare reimbursement structures, we believe that this research provides an encouraging and reassuring message to healthcare leaders as it justifies investing in hospital quality improvement projects. Specific recommendations regarding the exact allocation of resources are premature; however, there is broad support for action, and we suggest that this support will only continue to strengthen over time. 

## Figures and Tables

**Table 1 healthcare-11-02758-t001:** Descriptive statistics.

	Minimum	Maximum	Mean	Std. Deviation
** DEPENDENT VARIABLES **				
Net Patient Revenue per Discharge	−10,735.92	181,518.68	37,488.57	16,968.67
** INDEPENDENT VARIABLES OF INTEREST **				
HCAHPS Summary Star Rating	1	5	3.10	0.80
Hospital Compare Overall Rating	1	5	3.23	1.13
Readmission Rate	0.10	0.21	0.15	0.01
2021 Total Performance Score	6	92.67	33.78	11.4
2021 Clinical Domain Score	0	100	43.56	18.2
2021 Safety Domain Score	0	100	39.69	20.75
2021 Engagement Domain Score	5	100	31.72	17.03
2021 Efficiency Domain Score	0	100	20.19	25.00
** CONTROL VARIABLES **				
Total Assets per Bed (in millions)	−2.56	148.54	2.45	6.34
CC/MCC Rate	0.13	0.86	0.67	0.074
Rural	0	1	0.08	0.27
Market Concentration Index	0.02	1.00	0.35	0.33
Government Operated	0	1	0.13	0.34
For Profit	0	1	0.20	0.40
Average Daily Census	2.30	1953.80	146.36	169.94
Surgical CMI	1.01	5.33	2.98	0.56
Medical CMI	0.73	2.10	1.37	0.10
Case Mix Index	1.01	3.85	1.74	0.30
Payor Mix: Medicaid Days	0	0.79	0.09	0.08
Payor Mix: Medicare Days	0.00	0.79	0.28	0.10
Bed Utilization Rate	0.03	1.00	0.56	0.19
Average Age of Facility (in Years)	1.06	74.35	14.04	10.17
Average Length of Stay	0.80	15.70	4.88	1.04
Uncompensated Care (in millions)	0.13	1065.70	26.73	56.69
Charity Care (in millions)	0	722.20	10.11	26.00
Labor Compensation Ratio	−0.89	6.13	0.45	0.19
Southeast Region	0	1	0.30	0.46
Southwest Region	0	1	0.12	0.32
Midwest Region	0	1	0.24	0.43
West Region	0	1	0.18	0.38
Northeast Region	0	1	0.16	0.37

**Table 2 healthcare-11-02758-t002:** Regression results—HCAHPS, Hospital Compare, readmissions, and Total Performance Score.

Analysis of Net Patient Revenue Per Discharge (LN)	HCAHPS Summary Star Rating			Hospital Compare Rating			Readmission Rate			Total Performance Score		
	β	S.E.	Sig	β	S.E.	Sig	β	S.E.	Sig	β	S.E.	Sig
	N = 2096, Adj R^2^ = 48.2%			N = 2112, Adj R^2^ = 45.2%			N = 2126, Adj R^2^ = 43.9%			N = 2127, Adj R^2^ = 45.8%		
** INDEPENDENT VARIABLES **												
HCAHPS Summary Star Rating	0.088	0.007	***									
Hospital Compare Overall Rating				0.034	0.005	***						
Readmission Rate							−0.017	0.006	**			
2021 Total Performance Score										0.005	0.000	***
** CONTROL VARIABLES **												
Total Assets per Bed (in millions)	0.011	0.001	***	0.012	0.001	***	0.013	0.001	***	0.012	0.001	***
CC/MCC Rate	0.162	0.094	+	0.231	0.094	*	0.251	0.095	**	0.219	0.093	*
Rural	0.052	0.024	*	0.052	0.024	*	0.063	0.024	**	0.052	0.024	*
Market Concentration Index	0.111	0.017	***	0.141	0.017	***	0.128	0.018	***	0.138	0.017	***
Government Operated	0.055	0.016	***	0.078	0.017	***	0.074	0.017	***	0.081	0.016	***
For Profit	−0.088	0.014	***	−0.116	0.014	***	−0.129	0.014	***	−0.128	0.014	***
Average Daily Census	0.000	0.000	**	0.000	0.000	**	0.000	0.000	**	0.000	0.000	-
Surgical CMI	−0.185	0.018	***	−0.206	0.017	***	−0.218	0.018	***	−0.198	0.017	***
Medical CMI	−0.243	0.076	***	−0.341	0.076	***	−0.417	0.076	***	−0.349	0.075	***
Case Mix Index	0.424	0.032	***	0.506	0.031	***	0.524	0.032	***	0.502	0.031	***
Payor Mix: Medicaid Days	−0.001	0.001	-	−0.001	0.001	-	−0.001	0.001	*	−0.001	0.001	-
Payor Mix: Medicare Days	0.002	0.001	***	0.002	0.001	***	0.003	0.001	***	0.002	0.001	***
Bed Utilization Rate	−0.003	0.000	***	−0.004	0.000	***	−0.004	0.000	***	−0.004	0.000	***
Average Age of Facility (Years)	−0.001	0.001	**	−0.002	0.001	***	−0.002	0.001	***	−0.002	0.001	***
Average Length of Stay	0.095	0.006	***	0.091	0.006	***	0.085	0.006	***	0.090	0.006	***
Uncompensated Care (in millions)	0.001	0.000	***	0.001	0.000	***	0.001	0.000	***	0.001	0.000	***
Charity Care (in millions)	−0.001	0.000	*	−0.001	0.000	*	−0.001	0.000	*	−0.001	0.000	*
Labor Compensation Ratio	−0.002	0.000	***	−0.002	0.000	***	−0.003	0.000	***	−0.003	0.000	***
Southeast Region	−0.162	0.017	***	−0.163	0.017	***	−0.170	0.017	***	−0.165	0.017	***
Southwest Region	−0.157	0.021	***	−0.163	0.021	***	−0.165	0.021	***	−0.158	0.021	***
Midwest Region	0.013	0.016	-	0.017	0.017	-	0.015	0.017	-	0.012	0.017	-
West Region	0.096	0.018	***	0.087	0.019	***	0.089	0.019	***	0.080	0.019	***

Note: + *p* < 0.1, * *p* < 0.05; ** *p* < 0.01; *** *p* < 0.001; Northeast Region is the referent region.

**Table 3 healthcare-11-02758-t003:** Regression results—Hospital Value Based Purchasing sub-domains (clinical, safety, engagement, and efficiency).

Analysis of Net Patient Revenue Per Discharge (LN)	Clinical Outcomes Domain Score			Safety Domain Score			Engagement Domain Score			Efficiency Domain Score		
	β	S.E.	Sig	β	S.E.	Sig	β	S.E.	Sig	β	S.E.	Sig
	N = 2114, Adj R^2^ = 43.5%			N = 1820, Adj R^2^ = 46.2%			N = 2125, Adj R^2^ = 48.6%			N = 2126, Adj R^2^ = 46.3%		
** INDEPENDENT VARIABLES **												
2021 Clinical Domain Score	−0.001	0.001	-									
2021 Safety Domain Score				0.003	0.001	**						
2021 Engagement Domain Score							0.016	0.001	***			
2021 Efficiency Domain Score										0.008	0.001	***
** CONTROL VARIABLES **												
Total Assets per Bed (in millions)	0.012	0.001	***	0.011	0.001	***	0.011	0.001	***	0.013	0.001	***
CC/MCC Rate	0.363	0.098	***	0.017	0.106	-	0.314	0.091	***	0.177	0.093	+
Rural	0.064	0.025	**	0.032	0.028	-	0.040	0.023	+	0.043	0.024	+
Market Concentration Index	0.133	0.018	***	0.127	0.019	***	0.128	0.017	***	0.117	0.017	***
Government Operated	0.071	0.017	***	0.075	0.017	***	0.061	0.016	***	0.084	0.016	***
For Profit	−0.131	0.014	***	−0.145	0.014	***	−0.109	0.014	***	−0.127	0.014	***
Average Daily Census	0.000	0.000	**	0.000	0.000	***	0.000	0.000	***	0.000	0.000	***
Surgical CMI	−0.226	0.017	***	−0.223	0.019	***	−0.167	0.017	***	−0.199	0.017	***
Medical CMI	−0.491	0.079	***	−0.240	0.082	**	−0.295	0.074	***	−0.331	0.075	***
Case Mix Index	0.546	0.032	***	0.576	0.033	***	0.432	0.031	***	0.528	0.030	***
Payor Mix: Medicaid Days	−0.001	0.001	+	−0.001	0.001	-	−0.001	0.001	*	−0.001	0.001	*
Payor Mix: Medicare Days	0.002	0.001	***	0.003	0.001	***	0.002	0.001	***	0.003	0.001	***
Bed Utilization Rate	−0.004	0.000	***	−0.002	0.000	***	−0.003	0.000	***	−0.004	0.000	***
Average Age of Facility (Years)	−0.002	0.001	***	−0.002	0.001	**	−0.002	0.000	**	−0.002	0.001	***
Average Length of Stay	0.087	0.006	***	0.097	0.007	***	0.095	0.006	***	0.088	0.006	***
Uncompensated Care (in millions)	0.001	0.000	***	0.001	0.000	***	0.001	0.000	***	0.001	0.000	***
Charity Care (in millions)	−0.001	0.000	*	−0.001	0.000	+	−0.001	0.000	*	−0.001	0.000	*
Labor Compensation Ratio	−0.003	0.000	***	−0.003	0.000	***	−0.002	0.000	***	−0.003	0.000	***
Southeast Region	−0.168	0.018	***	−0.173	0.017	***	−0.176	0.017	***	−0.169	0.017	***
Southwest Region	−0.158	0.021	***	−0.175	0.022	***	−0.174	0.020	***	−0.155	0.021	***
Midwest Region	0.024	0.017	-	0.002	0.017	-	0.001	0.016	-	0.008	0.017	-
West Region	0.103	0.019	***	0.066	0.019	***	0.090	0.018	***	0.070	0.019	***

Note: + *p* < 0.1, * *p* < 0.05; ** *p* < 0.01; *** *p* < 0.001; Northeast Region is the referent region.

## Data Availability

All analyses were conducted in SPSS, Version 28, and all tables were constructed in Microsoft Excel.

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
