# Peer review of "An Exploratory Analysis of the Association between Hospital Quality Measures and Financial Performance"

_healthcare, 2023, doi:10.3390/healthcare11202758_

Round 1
Reviewer 1 Report
Dear Respectable Authors
Thank you for considering a significant area of research related to hospital quality measures and financial performance. As stated, you conducted an exploratory analysis to assess whether the quality of care from the perspective of the patient and clinical data are associated with a hospital’s ability to attract patients as measured by revenue. Your manuscript is well-written and is of interest but needs some minor revisions as follows;
- Title, based on what you stated in the aim of the study, your title does not adequately address what is stated in the objective.
- Abstract, please add your aim of the study in this section.
- Abstract, please add significance level for each variable.
- Abstract, which seven quality measures are significantly associated with improved net patient revenue per adjusted discharge?
- Line 106, the aim is better not to be directional. Please remove "positively".
- Line 274, what is your rationale for selecting these eight variables?
- Please refine your conclusion. The conclusion is a direct response to the aim of the study or your questions without statistical difficulties. Also, according to scientific writing standards, we are not allowed to use references in the conclusion section. Please remove the general statements give a direct answer to your hypotheses in this section and give one or two practical recommendations.
- Please add some details regarding the ethical code or ethical issues related to data acquisition.
- Discussion, considering that you have stated in the study method section that some data have been omitted (lines 257-260), does the absence of these data affect your results? Please discuss this.
Author Response
Reviewer #1
Comment #1: Title, based on what you stated in the aim of the study, your title does not adequately address what is stated in the objective.
Response #1: With respect, we are not sure what is meant by this comment. Can you clarify further?
______________________
Comment #2: Abstract, please add your aim of the study in this section.
Response #2: Thank you for your comment. We reworded the opening lines of the abstract (lines 11- 15) to read, “Hospitals are perpetually challenged by concurrently improving the quality of healthcare and maintaining financial solvency. Both issues are among the top concerns for hospital executives across the United States, yet some have questioned if the efforts to enhance quality are financially sustainable. Thus, the aim of this study is to examine if efforts to improve quality in the hospital setting have a corresponding association with hospital profitability.”
______________________
Comment #3: Abstract, please add significance level for each variable.
Response #3: We revised the abstract (lines 18-24) to read, “Using multivariate regression, we found that improving quality was significantly associated with our targeted measure of hospital profitability: the net patient revenue per adjusted discharge. Significant findings were reported for seven of eight quality measures tested, including the HCAHPS Summary Star Rating (p < 0.001), Hospital Compare Overall Rating (p < 0.001), All-Cause hospital-wide Readmission Rate (p < 0.01), Total Performance Score (p < 0.001), Safety Domain Score (p < 0.01), Person and Community Engagement Domain Score (p < 0.001), and the Efficiency and Cost Reduction Score (p < 0.001).”
______________________
Comment #4: Abstract, which seven quality measures are significantly associated with improved net patient revenue per adjusted discharge?
Response #4: We believe we addressed this concern with the revision in response to Comment #3.
______________________
Comment #5: Line 106, the aim is better not to be directional. Please remove "positively".
Response #5: The paper (lines 102-104) has been revised to read, “We specifically aim to assess whether the quality of care from the perspective of the patient and clinical data is associated with a hospital’s ability to attract patients as measured by revenue.”
______________________
Comment #6: Line 274, what is your rationale for selecting these eight variables?
Response #6: We have revised the draft to read (lines 96 – 102): “Thus, building on the work of the referenced studies and noting their recognized limitations, the research team seeks to advance our understanding of the association between several established quality measures and hospital profitability in acute care hospitals in the United States.”
We also added the following on lines 250-252: “Numerous quality assessment measures of acute care hospital operations are available from various sources. This facilitated our analysis and allowed us to choose a diverse set of independent variables drawn from well-established and publicly available data.”
______________________
Comment #7: Please refine your conclusion. The conclusion is a direct response to the aim of the study or your questions without statistical difficulties. Also, according to scientific writing standards, we are not allowed to use references in the conclusion section. Please remove the general statements give a direct answer to your hypotheses in this section and give one or two practical recommendations.
Response #7: We have moved the reference from the conclusion up to Section 2.3 “Hospital Quality of Care & Financial Performance” (lines 176-178). However, we do not concur that the conclusion is the place to identify which hypotheses were supported. We have made that clear in our “4. Results” section of the paper. Practical recommendations have been delineated in section “5.1 Practice Implications”.
______________________
Comment #8: Please add some details regarding the ethical code or ethical issues related to data acquisition.
Response #8: The following content was added to the “Institutional Review Board Statement” (lines 531-536): “Prior research of this type has been reviewed by the Texas State University Research Integrity and Compliance (RIC). According to the provisions in 45 CFR § 46.102 pertaining to "human subject" research, the RIC has previously determined studies of this type exclusively involve the examination of data originally collected and created by Definitive Healthcare, which provides data that is anonymous and publicly available. Therefore, the RIC has concluded this type of research does not use human subjects and is not regulated by the provisions in 45 CFR § 46.102 and therefore an IRB review of the study has not been required. “
______________________
Comment #9: Discussion, considering that you have stated in the study method section that some data have been omitted (lines 257-260), does the absence of these data affect your results? Please discuss this.
Response #9: Thank you for this comment. We have included reference to this issue as a limitation in our current draft (lines 475 – 480): “Several limitations are present in our study. First, the current study is drawn from a single data year (2022), and we have lagged independent and control variables (2021) to address endogeneity and reverse causality. In the future, we could consider the tested relationship by using longitudinal data and/or incorporating a more robust data set with more completed data. Although the current data set constitutes over half of acute care hospitals in the United States, our results might be altered if data were more comprehensive and complete for a larger number of hospitals.”
Reviewer 2 Report
This research aims to assess whether the quality of care from the perspective of the patient and clinical data are positively associated with a hospital’s ability to attract patients as measured by revenue.
The abstract is structured.
The introduction presents a good introduction to the topic, but along with the highlighted aspects, the quality should also be perceived through the lens of labor practices of the medical staff.
In this sense, in the literature exploration, reference should be made to recent studies that have developed quality assurance systems embedded in the sustainability of hospitals such as: Moldovan F, Moldovan L, Bataga T. Assessment of Labor Practices in Healthcare Using an Innovatory Framework for Sustainability. Medicina. 2023; 59(4):796. https://doi.org/10.3390/medicina59040796
At the end of the introduction, the objective of the research is presented and the hypotheses are formulated.
The research methodology is clearly described, but it would be desirable to specify the period in which the data were collected and the means used.
Also, the relevance of the volume of data in the sample should be emphasized.
The abbreviated notation of the variables next to the extended names could facilitate a better tracking of the content.
The results are clearly presented and are discussed in relation to other achievements in the literature.
Future research directions are presented.
The conclusions are concise and the bibliography is extensive and mainly well written with some small inconsistencies ex poz. 43.
Other minor editing errors
line 13 ”sustainable. led”,
line 34 ”outcomes According”, etc.
Author Response
Reviewer #2
Comment #1: The introduction presents a good introduction to the topic, but along with the highlighted aspects, the quality should also be perceived through the lens of labor practices of the medical staff.
In this sense, in the literature exploration, reference should be made to recent studies that have developed quality assurance systems embedded in the sustainability of hospitals such as: Moldovan F, Moldovan L, Bataga T. Assessment of Labor Practices in Healthcare Using an Innovatory Framework for Sustainability. Medicina. 2023;59(4):796. https://doi.org/10.3390/medicina59040796
Response #1: Thank you for your comment and we appreciate the viewpoint. To this end, this is why we included “Labor Compensation Ratio” in our analysis as a control variable. If unfamiliar with the LCR, it is calculated as noted below:
Labor Cost Ratio = (Total Salaries + Total Contract Labor Cost + Total Fringe Benefits) / Net Patient Revenue
We draw the inclination to use this variable as a control measure based on recent research. We believe the LCR captures a great deal pertaining to the labor practices of the medical staff – or at least the labor costs associated. See:
Beauvais, B., Kruse, C. S., Ramamonjiarivelo, Z., Pradhan, R., Sen, K., & Fulton, L. (2023). An Exploratory Analysis of the Association Between Hospital Labor Costs and the Quality of Care. Risk Management and Healthcare Policy, 1075-1091.
--------------------------
Comment #2: The research methodology is clearly described, but it would be desirable to specify the period in which the data were collected and the means used.
Response #2: As noted in lines 229 – 240, the data for the dependent variable, net patient revenue per discharge, was collected from the calendar year 2022 and obtained from Definitive Healthcare via publicly available Medicare Cost Reports. Data for the independent quality variables was obtained from the same source via publicly available data sources such as Hospital Compare, the Hospital Value Based Purchasing Program, and others. We’ve inserted a mention of the HVBP program in the updated draft (line 234). We hope this addresses your concern?
--------------------------
Comment #3: Also, the relevance of the volume of data in the sample should be emphasized.
Response #3: With respect, we are not sure what is meant by this comment. Can you clarify further?
--------------------------
Comment #4: The abbreviated notation of the variables next to the extended names could facilitate a better tracking of the content.
Response #4: Thank you for this note. However, all names are provided in extended format in the tables with the exception of the CC/MCC rate. This variable is already defined in extended and abbreviated format on line 305. If there are others we are missing, or that remain unclear, please let us know.
--------------------------
Comment #5: The conclusions are concise and the bibliography is extensive and mainly well written with some small inconsistencies ex poz. 43.
Response #5: Thank you. We will review the bibliography prior to resubmission.
--------------------------
Comment #6: Other minor editing errors…line 13 ”sustainable. led”, line 34 ”outcomes According”, etc.
Response #6: Thank you. We will review the draft again for spelling, grammar, and other errors prior to resubmission.
Reviewer 3 Report
The article 'An Exploratory Analysis of the Association Between Hospital Quality Measures and Financial Performance' addresses a research topic that was oriented towards the important problem of assessing the economic justification of efforts to improve the quality of services provided by hospitals, in light of the requirements of simultaneously improving the quality of healthcare and maintaining financial solvency. For the purposes of the research, the authors used a multivariate regression tool to assess whether multiple measures of quality are related to the adopted measure of hospital profitability: net patient revenue per adjusted discharge. This is a very important and interesting topic, worthy of scientific recognition.
The abstract defines the aim of the study and outlines the research direction with the methodology, and cites the main findings. I believe that the research gap in relation to existing studies should be more strongly defined in the abstract to justify the purpose of the study and to indicate the value of the study, its novelty.
An introductory section divided into two parts presents the background to the research, the scope of the research and the expectations set for it. A research gap is indicated, justifying the importance of the topic adopted. This point (scope 1.1. Background) needs to be elaborated. When indicating the level of spending in the USA compared to other countries, financial ranges should be given (make a comparison) or such statements should be dropped so that the content does not take on a vague character (line 29-30).
The point of a thematically divided literature review is very valuable. However, I believe that the introduction of this review is worth supplementing with a contemporary approach to sustainable management in order to justify, in the following section, the balancing of efforts to improve the quality of hospitals' performance and the financial results they achieve. The discussion of general orientations (universal nature) is then worth transferring to the hospital setting and further considerations. In this respect, it is worth, for example, studying the study https://doi.org/10.3390/su15118889. The remainder of the review highlights the gap in the literature and justifies more strongly the need for the research undertaken.
The hypotheses identified in section 2.4. "Conceptual Framework and Hypotheses" should not be presented in the literature review section. I suggest supplementing the methodology section with the research model of this article, presenting the hypotheses and correlating the individual research steps with the methods adopted and their overall presentation. In this respect, I believe that point 3 'Methods' should be refined. In addition, the research limitations should be more clearly defined (to relate their impact to the limitation of the research results in point 6 "Limitations and Suggestions for Future Research").
Item 4 "Results" - under the table (lines 366, 417) the text should be completed. Presentation of data very factual, albeit difficult to read. It might be worth reinforcing this part of the paper with a small visualisation to make the content easier to perceive (just a suggestion for the Authors to consider).
In section 5, 'Discussion', it may be worth strengthening the implications (section 5.1).
Section 7 'Conclusions' needs refinement. Reference should be made to the purpose of the research and the results should be presented in the context of responding to the diagnosed research gap, emphasising the significance of the article - its novelty.
This is a good, interesting article. Before publication, however, it needs refinement according to the suggestions indicated in the review.
Author Response
Reviewer #3
Comment #1: The abstract defines the aim of the study and outlines the research direction with the methodology, and cites the main findings. I believe that the research gap in relation to existing studies should be more strongly defined in the abstract to justify the purpose of the study and to indicate the value of the study, its novelty.
Response #1: Thank you for this comment. We have included the following line in the abstract of the current draft (lines 16-18): “Recent and directly relevant research on this topic is very limited leaving practitioners uncertain about the wisdom of their investments into quality and patient safety enhancing interventions.”
------------------------
Comment #2: An introductory section divided into two parts presents the background to the research, the scope of the research and the expectations set for it. A research gap is indicated, justifying the importance of the topic adopted. This point (scope 1.1. Background) needs to be elaborated. When indicating the level of spending in the USA compared to other countries, financial ranges should be given (make a comparison) or such statements should be dropped so that the content does not take on a vague character (line 29-30).
Response #2: We appreciate this clarification. We have added the following line (lines 37-39) to our work, “As of 2021, the percent of GDP spent on health in the US stood at 17.8%, far higher than the next highest group of developed nations including Germany (12.8%), France (12.4%), Great Britain (11.9%) and Canada (11.7%)”
------------------------
Comment #3: The point of a thematically divided literature review is very valuable. However, I believe that the introduction of this review is worth supplementing with a contemporary approach to sustainable management in order to justify, in the following section, the balancing of efforts to improve the quality of hospitals' performance and the financial results they achieve. The discussion of general orientations (universal nature) is then worth transferring to the hospital setting and further considerations. In this respect, it is worth, for example, studying the study https://doi.org/10.3390/su15118889. The remainder of the review highlights the gap in the literature and justifies more strongly the need for the research undertaken.
Response #3: Thank you for this suggestion. However, we did not perceive that inclusion of this vein of management literature would be supportive of our current work. However, we did see how this might be a worthwhile pathway to consider in future research.
------------------------
Comment #4: The hypotheses identified in section 2.4. "Conceptual Framework and Hypotheses" should not be presented in the literature review section. I suggest supplementing the methodology section with the research model of this article, presenting the hypotheses and correlating the individual research steps with the methods adopted and their overall presentation. In this respect, I believe that point 3 'Methods' should be refined. In addition, the research limitations should be more clearly defined (to relate their impact to the limitation of the research results in point 6 "Limitations and Suggestions for Future Research").
Response #4: Per your suggestion, we have moved the conceptual framework into the Methods section of the paper, although in our experience the conceptual framework comes at the end of the literature review as it is informed by the preceding literature content in the paper. We have also added content in section 3.4 “Independent Variables of Interest” pertaining to which variables support each tested hypothesis (lines 266-267, 274-275, 279-280, 297-299).
------------------------
Comment #5: Item 4 "Results" - under the table (lines 366, 417) the text should be completed. Presentation of data very factual, albeit difficult to read. It might be worth reinforcing this part of the paper with a small visualisation to make the content easier to perceive (just a suggestion for the Authors to consider).
Response #5: We have updated both Table 2 & Table 3. Regarding a visualization…thanks for the suggestion. We struggled with how to do this clearly and could not come up with anything more effective than presentation in a basic table format. We will work with the MDPI copyedit team to ensure these figures are as clear as possible.
------------------------
Comment #6: In section 5, 'Discussion', it may be worth strengthening the implications (section 5.1).
Response #6: We have updated our current draft to read (lines 469-482), “Others have previously suggested that improved hospital quality (e.g., surgical care improvement, safety scores, etc.) was associated with improved financial performance [41, 42]. Prior researchers exposed some of the interactions between quality and profitability in hospital systems. However, in this study, we sought to take a more expansive look into the organizational data to see if other quality metrics are associated with financial performance in the same way prior researchers have found with a narrower look at strictly patient safety performance. And, overall, our results were highly consistent within our own study and with prior authors’ findings. Specifically, we can infer from our findings that many of the steps taken to improve patient perceptions of quality and safety, reduce readmissions, and improve efficiency are all in the best interest of the patient but also serve to financially support the organizations long term economic viability. Although we believe more work needs to be done in this area, we contend that our findings continue to add to the evidence base that affirms the business case for improving organizational quality. With our results, healthcare leaders' support of focused investments in quality and efficiency can be profitable irrespective of the influence of regulatory or value-based purchasing initiatives.”
------------------------
Comment #7: Section 7 'Conclusions' needs refinement. Reference should be made to the purpose of the research and the results should be presented in the context of responding to the diagnosed research gap, emphasising the significance of the article - its novelty.
Response #7: We have updated the current draft to read, “Historically hospital leaders have been reluctant to invest in quality improvement and safety programs. We contend those days are in the past. There has been, and likely always will be, a perceived trade-off between quality improvement and hospital profitability. To our knowledge, this association has never been studied in granular detail with a contemporary set of quality and safety measures. So, in our research, we called the quality-profitability tradeoff premise into question and examined the association between eight unique health care quality variables and hospital profitability. Based on our research, we can confidently indicate that effort expended to enhance the patient experience, reduce readmissions, and improve patient safety is associated with improved hospital financial performance. Given the expanding role of value-based case incentives and evolving healthcare reimbursement structures, we believe that this research provides an encouraging and reassuring message to healthcare leaders as it justifies investing in hospital quality improvement projects. Specific recommendations regarding the exact allocation of resources are premature, however, there is broad support for action, and we suggest that this support will only continue to strengthen over time.”
Reviewer 4 Report
The calculations for the central hypotheses in 240-247 are presented well.
The discussion of the lower part of Tables 2 and 3, the regression coefficients related to Facility Attributes, was a little confusing for me. Is this regression with individual ratings (Table 2) and domain scores (Table 3), as implied in the tabular structure? Or is the regression with per patient revenue, as for example implied by lines 424-428? Or are the column headings the independent variable controlled for? For example, line 424 talks about a 1.2%, or beta=.012 for total assets per staffed bed, but the beta in the last column, under Efficiency domain score, on the same row is .013. I think just explaining the regression model in a little bit of detail would help.
I also think the multivariate model could be used to delineate some interesting effects between variables. For example, are the negative betas for the south east region explained away by the rural/urban mix and market concentration etc?
Author Response
Reviewer #4
Comment #1: The discussion of the lower part of Tables 2 and 3, the regression coefficients related to Facility Attributes, was a little confusing for me. Is this regression with individual ratings (Table 2) and domain scores (Table 3), as implied in the tabular structure? Or is the regression with per patient revenue, as for example implied by lines 424-428? Or are the column headings the independent variable controlled for? For example, line 424 talks about a 1.2%, or beta=.012 for total assets per staffed bed, but the beta in the last column, under Efficiency domain score, on the same row is .013. I think just explaining the regression model in a little bit of detail would help.
Response #1: Thanks for your comments. We have worked to clean up our tables in the revised draft. We also want to point out in section 4.2 “Secondary Findings” that we now state (line 402-403), “As a sample of the findings we will use the HCAHPS Summary Rating results from Table 2 as our reported analysis in this section.” Given there are eight regressions, we just didn’t think it was possible to cover all of these secondary findings, nor was it our primary focus, so we opted to simply report one independent variable’s results as a representation of the entire set.
--------------------------------------
Comment #2: I also think the multivariate model could be used to delineate some interesting effects between variables. For example, are the negative betas for the south east region explained away by the rural/urban mix and market concentration etc?
Response #2: Thanks for this suggestion. Although we didn’t include any interaction terms in the current analysis, this would be a great new line of research to undertake. As such, we have included a comment in section 6 “Limitations and Suggestions for Future Research” (line4 499 – 500) that reads, “Inclusion of interaction terms among our studied variables might also be an interesting addition to the research.”
Round 2
Reviewer 3 Report
The authors have revised the article in accordance with the review.